# Ethereum-Based Information System for Digital Higher Education Registry and Verification of Student Achievement Documents

Yerlan Kistaubayev [1], Galimkair Mutanov [2,*], Madina Mansurova [1], Zhanna Saxenbayeva [3] and Yassynzhan Shakan [1]

[1] Faculty of Information Technologies, Al-Farabi Kazakh National University, Almaty 050010, Kazakhstan
[2] Institute of Information and Computational Technologies, Committee of Science of the Ministry of Education and Science, Almaty 050010, Kazakhstan
[3] Department of Digital Transformation, International University of Information Technologies, Almaty 050010, Kazakhstan
[*] Correspondence: galimkairmutanov@gmail.com

**Abstract:** Blockchain is a new and modern technology that is gradually being used in various fields due to its ability to decentralize and organize secure and reliable data exchange and storage. One of the related research areas generating increasing interest is the field of education, with particular focus on the digitization and automation of educational management processes and the ability to store and verify digital documents about student progress. The main goal of this study is to develop a platform that creates a unified digital register of students' educational achievements, which is one of the most pressing issues in the field of education, based on the Ethereum blockchain architecture. Blockchain is expensive; therefore, there is a need to consider performance criteria when evaluating any decision made about the technology, especially the most important aspects such as predicting traffic behavior, estimating transaction costs and providing the necessary indicators of system quality and functionality. However, most research ignores the evaluation of performance indicators, such as throughput, the speed of transactions and the amount of data stored in the Ethereum blockchain database, which are the main evaluation criteria. This paper aims to eliminate this gap by evaluating the performance of the developed platform and by discussing the obtained experimental results. Thus, the main results of this work are the design and deployment of a blockchain platform and the analysis of its transaction costs. We conclude that the proposed blockchain solution is applicable as a system for the accounting and verification of loans and student academic achievements.

**Keywords:** blockchain; higher education; smart contract; evaluation; performance





## 1. Introduction

The processes of globalization, the transition to a digital economy and Industry 4.0, the emergence of new professions and the transition to a digital society have had a strong impact on the field of education. The pandemic period contributed to the transformation of education, significantly expanding the possibility of online learning and forcing people to change their field of activity, improve their digital skills and pursue additional education. With the support of international organizations, such as Erasmus+ and DAAD, the number of joint educational programs and academic mobility programs is increasing, which allows students to receive loans to attend various universities around the world [1,2]. All these processes lead to the need to transform the system of accounting for student credits and academic achievements into a flexible, reliable and decentralized digital form [3]. Blockchain-based developments have already proven themselves in various fields. The number of blockchain applications in the field of education has grown rapidly [4]. In addition to securely storing student records and their academic achievements [5], blockchain

solutions are used to account for credits received as part of lifelong learning, the accumulation of digital badges and the confirmation of micro-qualifications [6]; this serves as a connective infrastructure between formal and informal learning [7]. Moreover, note that a key document that testifies student academic performance and that confirms previous achievements is an academic transcript with a list of completed disciplines and grades for them. For example, in higher educational institutions of the Republic of Kazakhstan, as in most universities across the world, transcripts are automatically generated after the end of the semester.

Verification of the authenticity of transcripts and the data contained in them is complicated by the fact that each university independently determines 80% of the structure and content of the curriculum. According to the standard rules of the activity of educational organizations of the corresponding types and of types approved by order of the Ministry of Education and Science of the Republic of Kazakhstan (MES RK), dated 30 October 2018 (No. 595), higher education institutions are granted academic freedom to form educational programs [8]. The MES RK has a unified higher education management system (UHEMS) with a register of the educational programs of higher and postgraduate education. UHEMS stores information about the goals, expected results and disciplines of educational programs of universities of the Republic of Kazakhstan [9]. The MES RK is responsible for determining the structure and content of general education disciplines. Each educational program undergoes a procedure of examination and approval in the register. Moreover, the MES RK evaluates each educational program for compliance with the expected stages of training. The procedures for evaluating programs and the requirements for awarding a bachelor's, master's or PhD degree are described in the State Mandatory Standard of Education (SMSE). The procedure for issuing a diploma and awarding qualifications is determined by each university independently and is reflected in the academic policy of the university. In general, this procedure includes the following steps:

1. The faculty send a memo to the university registrar's office, indicating the students who have fulfilled the requirements for obtaining qualifications.
2. The registrar's office checks whether the specified students have fulfilled all the requirements for obtaining a qualification.
3. The registrar's office makes a new entry in the registration book containing students' surnames, degrees, serial numbers their diplomas and other necessary information.
4. An employee of the registrar's office prints student diplomas on special forms.
5. At the request of the SSO, the chairman of the state commission signs diplomas.
6. The registrar's office sends the signed diplomas to the office of the rector of the university.
7. The head of the university signs the diplomas and returns them to the registrar's office.
8. Students are awarded diplomas at a graduation ceremony.

The workflow for the registration of the academic mobility of students or for transferring to another university is as follows:

1. A student applies to transfer to another university or to register for academic mobility.
2. The faculty and the registrar's office prepare an academic transcript of the student with a complete history of academic performance.
3. The head (the dean, vice-rector or rector—the university determines which one independently) and the registrar's office sign the transcript.
4. The registrar's office sends the transcript to the student.

Based on this description of the steps required in the document issuance process, it is easy to understand why problems may arise in the paper document flow, i.e., there is the possibility of technical error, fraud with the help of an unscrupulous employee and forgery of signatures. Therefore, to ensure the protection and integrity of data, blockchain technology is the best solution.

The purpose of this research is to develop and test a platform based on Ethereum technology for storing and verifying information about student academic performance. Maintaining the confidentiality of documents allows students to create a permanent public

record, to be protected from changes in the educational institution or from loss of personal data, to share it with employers and to monitor the level of their own academic achievements [10]. This paper is a continuation of our research on the use of blockchain technologies in education. Earlier, the architecture of the platform based on a globally distributed peer-to-peer network was presented; this allows for student registration and the authentication of educational documents [11]. During practical implementation of the platform, we realized that only a small number of papers evaluate the performance of blockchain solutions. Most relevant research papers are devoted to general blockchain problems, such as decentralization, transparency, security and data immutability. We define these criteria as theory-based solutions. Only a small number of papers present results on important factors such as time, data storage and transaction costs; these are considered the main computing resources. We define these characteristics as practicality based.

Based on the relevant scientific literature, the following results were obtained:

1.  Characteristic problems associated with the functioning of blockchain systems used in the field of education were identified.
2.  The architecture of the UniverCert blockchain system proposed in previous work was improved.
3.  Two evaluation criteria were formulated:
    (a)  Compliance of the developed prototype with the theoretical aspects of blockchain systems.
    (b)  Compliance of the developed prototype with performance requirements.
4.  A working prototype of the UniverCert blockchain system was developed.

This paper presents the UniverCert platform for higher education institutions, developed based on the hybrid architecture of the Ethereum blockchain, which allows for the creation of a single digital register of official documents with the ability to integrate data provided from external university platforms. Transcripts indicating academic achievements for each semester and a diploma for higher/postgraduate education are considered official documents of the learning process. The platform provides verification services for these documents for third-party organizations. This paper offers practical solutions in terms of time, latency and data storage; the theoretical analysis is supported by experimental results.

This paper proceeds as follows. Section 2 introduces related work in terms of three key aspects: time consumption, data storage and cost per transaction. Section 3 introduces the architecture of the UniverCert platform. Section 4 sets forth descriptors of the second criterion and experimental results in detail. Section 5 presents our conclusions.

## 2. Related Work

A review of related work revealed several common problems from using blockchain technology in the field of education, as shown in Table 1.

**Table 1.** Problems from using blockchain technology in the field of education.

| No. | Problems | References |
|:---:|:---|:---:|
| 1 | Eliminating data control by third parties | [12] |
| 2 | Ensuring security and non-tamperability (sharing) of data | [13–15] |
| 3 | Counterfeiting, loss and validation of academic records | [16,17] |
| 4 | Digitalization and poor management of the education system | [18,19] |

Our analysis shows that most published work is based on theoretical research and is limited to presenting the architecture of blockchain systems without highlighting the issues of practical implementation and the effectiveness of systems. The literature evaluates blockchain systems in terms of indicators such as decentralization, transparency, immutability and security. We believe that researchers focus on these indicators for several reasons. First, the aforementioned problems of blockchain systems in education are common in a wide range of blockchain applications in other sectors, such as logistics [20], energy [21],

business [22], healthcare [23] and tourism [24]. Second, approaches to solving problems are similar, differing only in the technologies and security protocols used. Third, the work reflects theoretical, well-studied aspects of blockchain construction. We propose combining these indicators and associating them with the first theory-based criterion for evaluating blockchain systems (Table 2).

**Table 2.** Solution category.

| Blockchain Systems/Key Factors | Theory-Based | | | | Practical | | |
|---|---|---|---|---|---|---|---|
| | Decentralization | Transparency | Immutability | Safety | Time | Data Volume | Transaction Cost |
| UZHBC | + | + | + | + | - | - | - |
| McRhys | + | + | + | + | - | - | - |
| ECVS | + | + | + | + | - | - | - |
| ECBC | + | + | + | + | + | + | - |
| Brazil proposal | + | + | + | + | - | - | + |
| World State | + | + | + | + | + | + | - |
| Ethna | + | + | + | + | + | - | - |
| UniverCert | + | + | + | + | + | + | + |

Furthermore, from the point of view of practical implementation, note that blockchain is one of the more expensive technologies for storing and synchronizing distributed registry data. Therefore, when implementing blockchain systems, performance indicators should be evaluated. There are numerous performance indicators, including time consumption, data storage, cost per transaction, scalability, throughput and latency [25]. Due to the importance of performance indicators and insufficient levels of research on them, we combined them into a second practicality-based criterion for evaluating blockchain systems. We selected the first three indicators for analysis.

Thus, the purpose of this work is to determine the criteria for evaluating both the effectiveness of a blockchain system and the results from computational experiments. Table 2 presents two criteria and the main indicators corresponding to them, according to which a comparative analysis of existing blockchain solutions is carried out.

Most existing research falls into the first category and covers all the main factors of the primary criteria [26,27]. However, research papers that take into account aspects of the second evaluation criteria are rare. Taking the Educational Certificate Blockchain (ECBC) as an example [28], this paper covers two very important factors of the second criteria: latency and data storage. CBC uses peer-to-peer collaboration to create blocks named MPT-Chain instead of competition. This leads to a reduction in transaction time and block size. First, a theoretical analysis of these two factors is carried out, followed by mathematical calculations and experimental results. In this paper, the time taken to create a block and the block size are compared with Bitcoin using a diagram. Statistics show that both parameters act as a much better indicator than Bitcoin. However, transaction cost, one of the most significant factors, is not provided.

The Brazilian university system has used blockchain and smart contracts in higher education institutions, such as universities and colleges [29]. This study highlights the need for a reliable and secure way to control the digitization of diplomas and academic credits in the field of education in Brazil. It lists several problems, such as duplication or errors related to paperwork, the possibility of fraud and failures in the existing system due to natural disasters. To avoid these problems, a blockchain-based solution is proposed that includes all the descriptors of the initial evaluation. The paper not only meets the requirements of the primary criteria but also covers one of the most important factors of the secondary criteria; more precisely, transaction costs are estimated on the Ethereum platform. The proposal uses three smart contracts and calculates the cost of each, followed by an estimate of the cost of launching the solution in Brazil, namely the amount of gas per student per year. However, this paper does not address two important aspects of the secondary criteria, namely data storage and latency.

In the World State study, a model is proposed that predicts the performance and storage of executable contracts for the Ethereum Consortium Blockchain based on transaction

volume [30]. This article focuses on the relationship between time consumption and data storage using formulas; then, an experiment is conducted to confirm the predicted data. Thus, the authors consider two estimated indicators, namely time consumption and storage. However, the cost of transactions is not taken into account.

Ethna [31] provides a tool called the Ethereal network analyzer for research and analysis of the Ethereum blockchain P2P network. Although the main purpose of this paper is to describe a new method to accurately measure the degrees of Ethereum nodes, the authors also measure the transaction–propagation time, a secondary evaluation criterion.

Our proposal differs from the above solutions. In particular, our previous study covers all aspects of the first evaluation criteria, whereas the present study discusses practical solutions in terms of latency, data storage and transaction costs, which are key factors of the secondary criteria. Further, our hypothesis is confirmed by the results of the experiment. In addition to the cost and quantity of transactions, we also take into account the speed of the network and execution of the transaction, thereby conducting an extended evaluation of the platform.

Currently, Ethereum is the largest platform for running decentralized applications; however, it has certain limitations that cannot be solved now. First, these are low throughput and high gas fees. There are more optimized and younger second layer networks, such as Solana and Polygon. The main goals of these networks are to reduce the amount of time it takes for blockchain nodes to synchronize or reach a consensus.

The Solana network uses its own Proof of History (PoH) algorithm along with the Proof of Stake (PoS) algorithm. PoH allows one to record an event that has occurred even before the information about the transaction is added to the blockchain. In this way, nodes on the Solana blockchain can create blocks without having to "coordinate" their actions with the entire network. This reduces the amount of transaction time. It takes 400 ms to generate one block in Solana. The maximum network throughput exceeds 50,000 TPS. However, there are only a few dozen decentralized applications (dapps) running on the network today, whereas Ethereum already has several thousand. Moreover, during the operation of the network, several network failures have been recorded due to hacks and incorrect algorithms.

The Polygon network is a second-level blockchain and is compatible with all EVM standards, with transaction speeds of up to 7000 per second. The blockchain itself is flexible and can be customized to the needs of each project. The average cost of a transaction is 0.002 USD. The network does not have miners; however, it does have validators and delegates. Validators stake their tokens as collateral and become part of the consensus mechanism. Delegates pass their coins to the validator and profit from staking. The risk of using the network is its age and the presence of other competitive analogs.

In general, the goals of the listed new networks can solve the shortcomings of our solution. However, it is necessary to consider the following risks: these networks have undergone limited testing with a sufficient number of users; there has been unstable network operation during their existence; there is a lack of other scientific research on the application of networks in education; and there is a lack of experience of authors within these networks.

We chose to use the Ethereum consortium. The network is a consortium and is non-public. Therefore, questions about transaction execution time and throughput are regulated. In addition, the cost of a transaction is not a problem for such a network, because the participants are not interested in earning but in the life of the network.

## 3. UniverCert Proposal

This section presents a blockchain-based solution for securely storing the history of the academic performance of students and graduates to create official documentation of the educational process. Academic performance history is stored in the format of the studied courses, with the grades received in the form of a transcript. This format is universal and corresponds to the fact that it is possible to store the academic track record of a person at

all stages of education, starting from the initial phase and ending with advanced training courses at the stage of a professional career. This solution provides for participation with three types of participants, as shown in Figure 1.

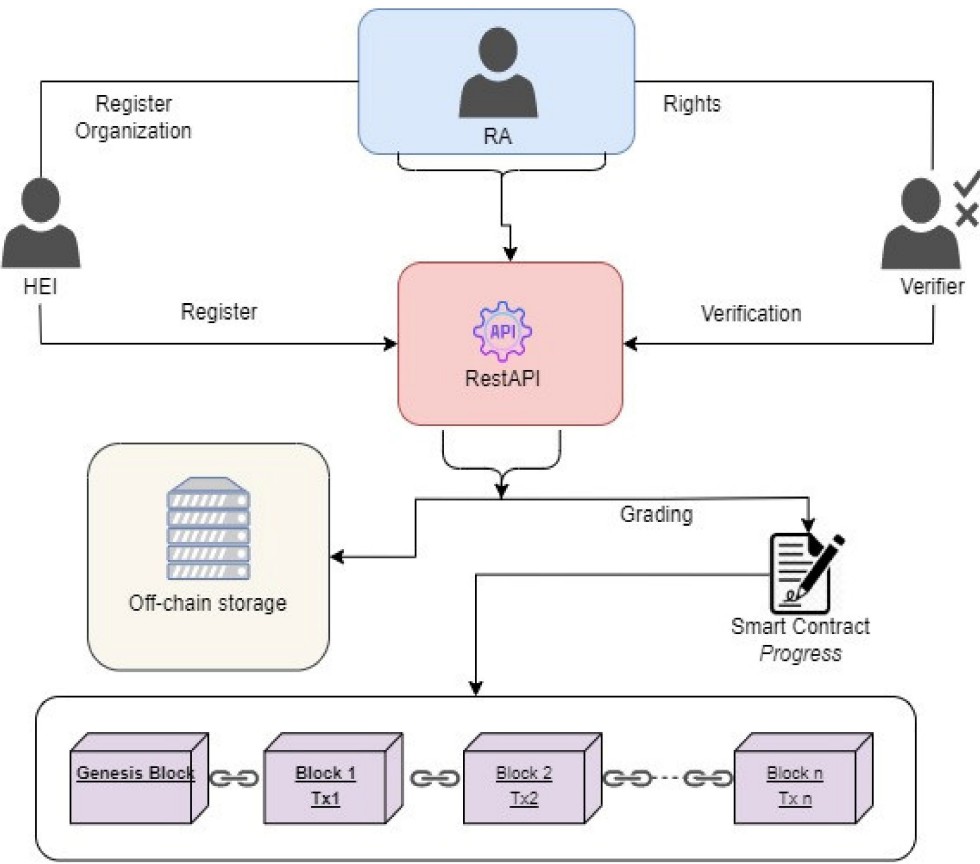

**Figure 1.** Architecture of the proposed platform.

The first is the role administrator (RA). This participant is responsible for registering an organization (Higher Education Institution [HEI], Verifier) in the platform and for providing it with the necessary access to the Rest API, through which all transactions are made. In addition, the RA is the owner of an off-chain database where reference information about participants, students and graduates is stored.

The second is the HEI, the main participant and user of the UniverCert platform, which makes transactions to store data on the academic performance of students and graduates. In addition to making a transaction in the blockchain database, this participant actively uses off-chain storage to store reference information and synchronize it with the database of learning management systems of educational institutions.

The third is the Verifier, a platform participant who performs operations to verify the authenticity of the academic achievements of a student or graduate by obtaining a digital version of educational documents. These participants include employers, recruiters and other educational institutions. As an example, an employer can obtain a reliable educational document (diploma supplement) of the applicant and can analyze it for compliance with the position. Another example is an educational institution to which a student seeking academic mobility is applying; in this case, it is important for the educational institution to have reliable information about the student's courses to correctly compile and understand the student's individual educational progress.

In a previous paper, a prototype with two smart contracts was proposed: Smart contract-1 for storing information about students and Smart contract-2 for registering their progress. However, during practical implementation of the platform, it became

clear that the use of two smart contracts does not meet the basic requirements of the second evaluation criteria. Therefore, only Smart contract-2 was retained.

According to the old prototype, Smart contract-1 processed the following data about a student: university, specialty, year of admission and graduation, level of study and academic degree. These data in the original form should have been stored in the blockchain database, and they can be obtained by a student ID without additional access. The process of making a transaction for registering a student in the blockchain database is carried out by linking the student to the specialty of a particular university. Moreover, all stored information consists entirely of textual information (including the name of the specialty, qualification and university in different languages), which affects the cost of the transaction.

To optimize costs, the off-chain database can be used to save all textual content. We use the identifiers of this database when making a transaction in the blockchain. In this case, the student's registration data are stored mainly in an off-chain database, and there is no need to duplicate the information in the blockchain database. Moreover, the student's registration information does not represent critical information that requires resorting to blockchain technology.

The interface between the participants and the blockchain is implemented by a single smart contract created by the RA. This smart contract is called "Progress," and it guarantees that a university with the appropriate access level in the platform can record events only in relation to its students. To correctly identify the university, the RA creates special accounts in the off-chain database to connect to RestAPI, as well as in the blockchain platform for making a transaction.

Figure 2 shows the step-by-step algorithm of the work and actions of participants and the role of the platform. Verification of the participant's rights to perform an operation on the platform is determined at the off-chain database level.

To save resources in making a transaction on the blockchain database and to comply with rules for storing personal data, the student's academic performance contains student/graduate IDs, educational programs and course IDs with grades. More detailed information on the students, graduates, specialties and courses can be found in the off-chain database.

Services in the RestAPI are implemented in such a way that, when making a transaction in the blockchain database, a student must first register all necessary reference information in the off-chain database and obtain the necessary identifiers.

### 3.1. Improved Architecture of the UniverCert Platform

This section describes the implementation of our solution. To create the smart contract mentioned in Section 3, the contract-oriented programming language Solidity and the Ethereum blockchain platform are used. The diagram and tables in the following subsections show a more detailed description of the working version of the platform and an analysis of the output data obtained using Remix IDE.53. These subsections contain key information on the test transactions. The prototype source code is available at https://gitlab.com/p9239/uni_cert/-/tree/main/, accessed on 22 October 2022.

Figure 3 shows the architecture of the UniverCert platform, which includes the following subsystems: web application, RestAPI layer, off-chain storage and blockchain storage.

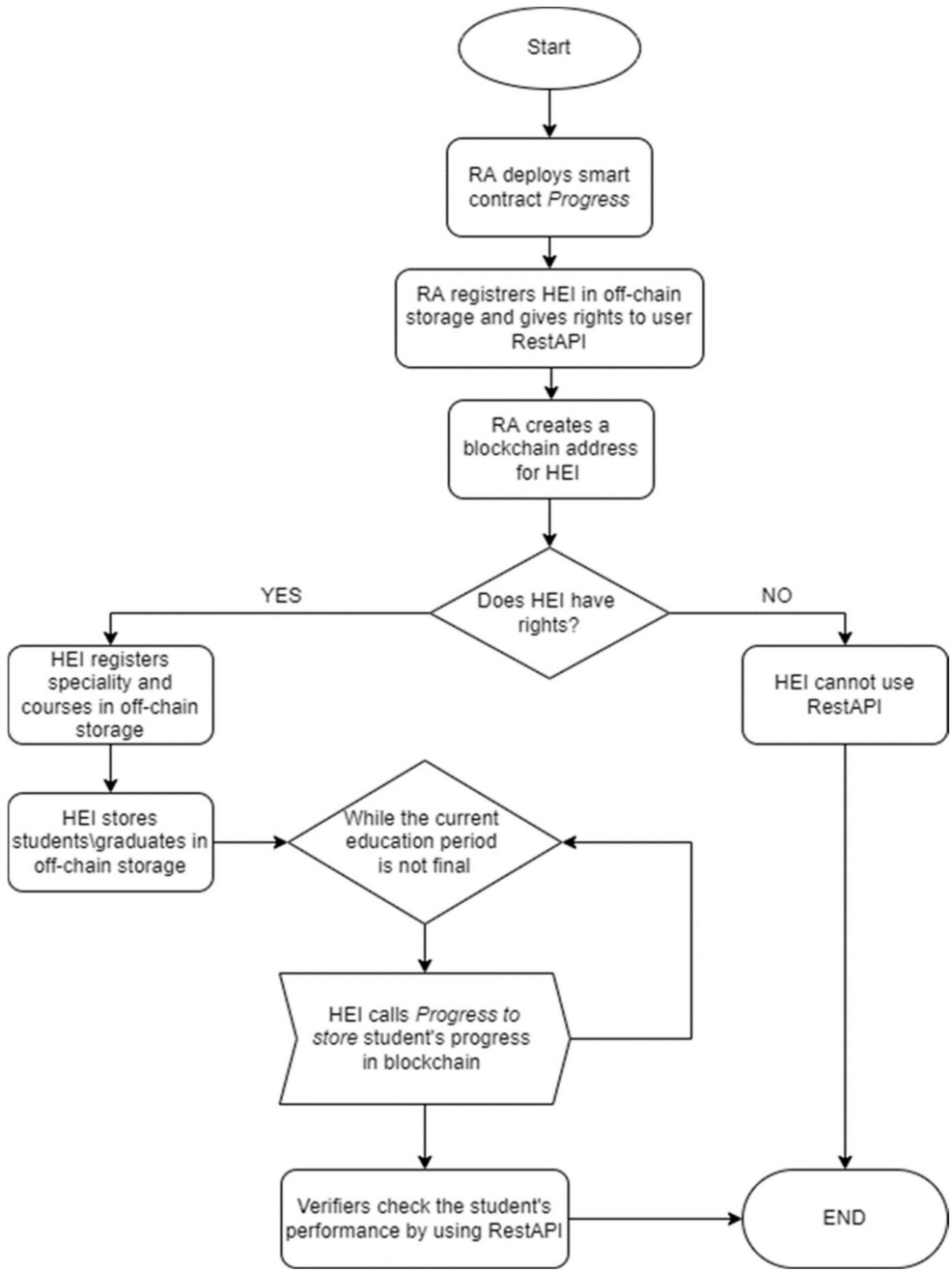

**Figure 2.** Platform workflow.

### 3.2. Web Application Layer

The web application is a subsystem for managing the access of HEI and Verifier participants on the UniverCert platform. In the first stage, the HEI, or the Verifier (e.g., recruiter or employer) submits an application for access to the platform. The RA examines the application and decides whether to accept the application. If the answer is positive, the applicant receives an e-mail with instructions on using the RestAPI, as well as credentials for connecting to the web application and the RestAPI. The web application also provides an opportunity to search for students and graduates, to get acquainted with their academic achievements and to download soft versions of transcripts of academic performance.

### 3.3. RestAPI Level

RestAPI (representative state transfer): Access to the blockchain platform is realized through the RestAPI channel. RestAPI stores readily-made methods for storing and searching data using filters (student and university ID). Data modification methods in the off-chain and blockchain databases are available to the RUSH participant, and data reading is also available to the Verifier.

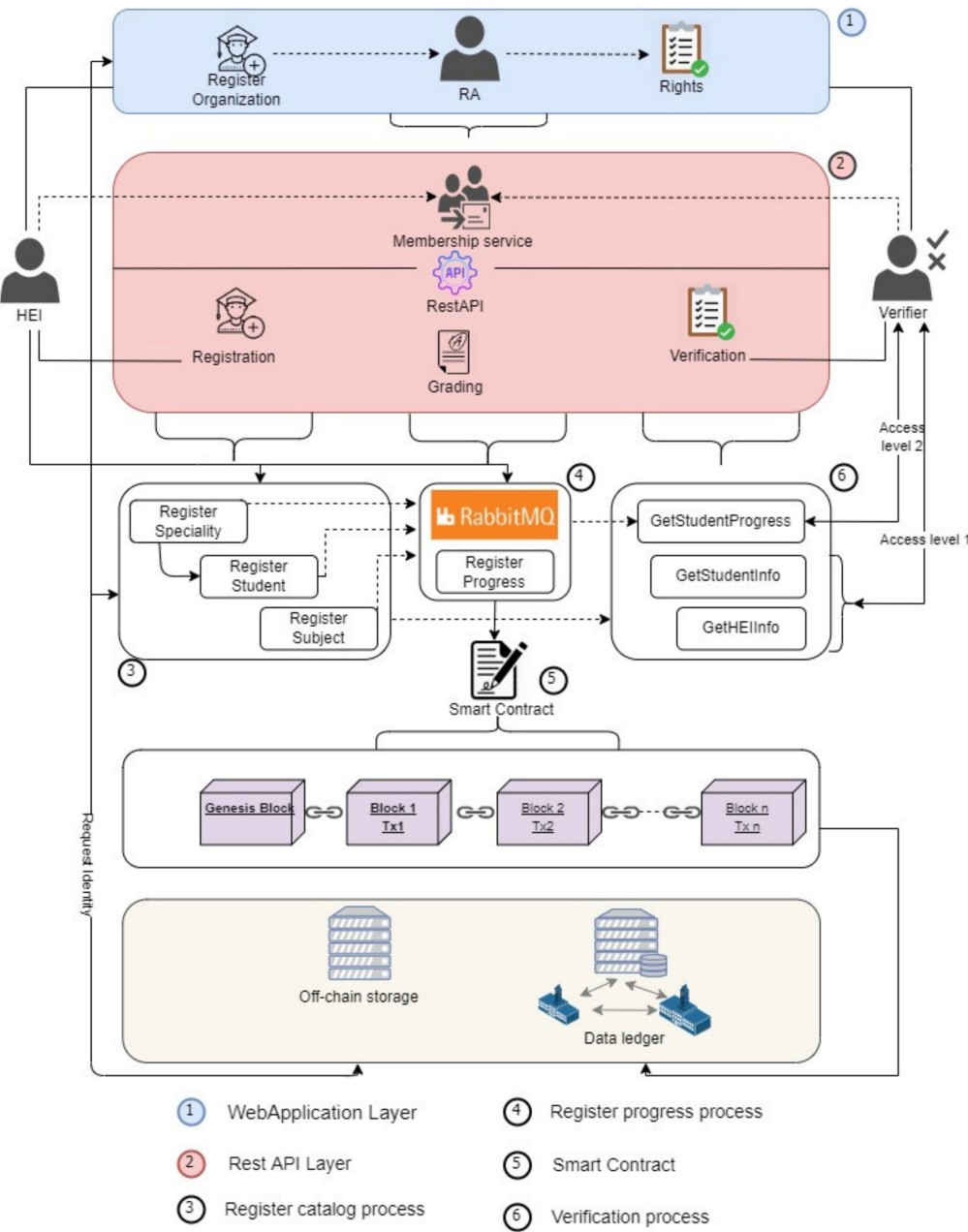

**Figure 3.** Detailed architecture of the UniverCert platform.

RestAPI organizes work with two databases: off-chain and blockchain databases. Accordingly, methods in RestAPI are divided into three categories: methods of the identification level (Identity), methods of registering reference information in an off-chain database (OffChain) and methods of making a transaction with a blockchain database (Blockchain).

The Identity category methods are designed to authorize a participant and to obtain a Jwt token for the further use of RestAPI methods, as well as for the registration and confirmation of participants at the web application level.

The Off-Chain category methods are implemented to work with off-chain database tables. The architecture of this database is shown in Figure 4, which clearly shows all the tables.

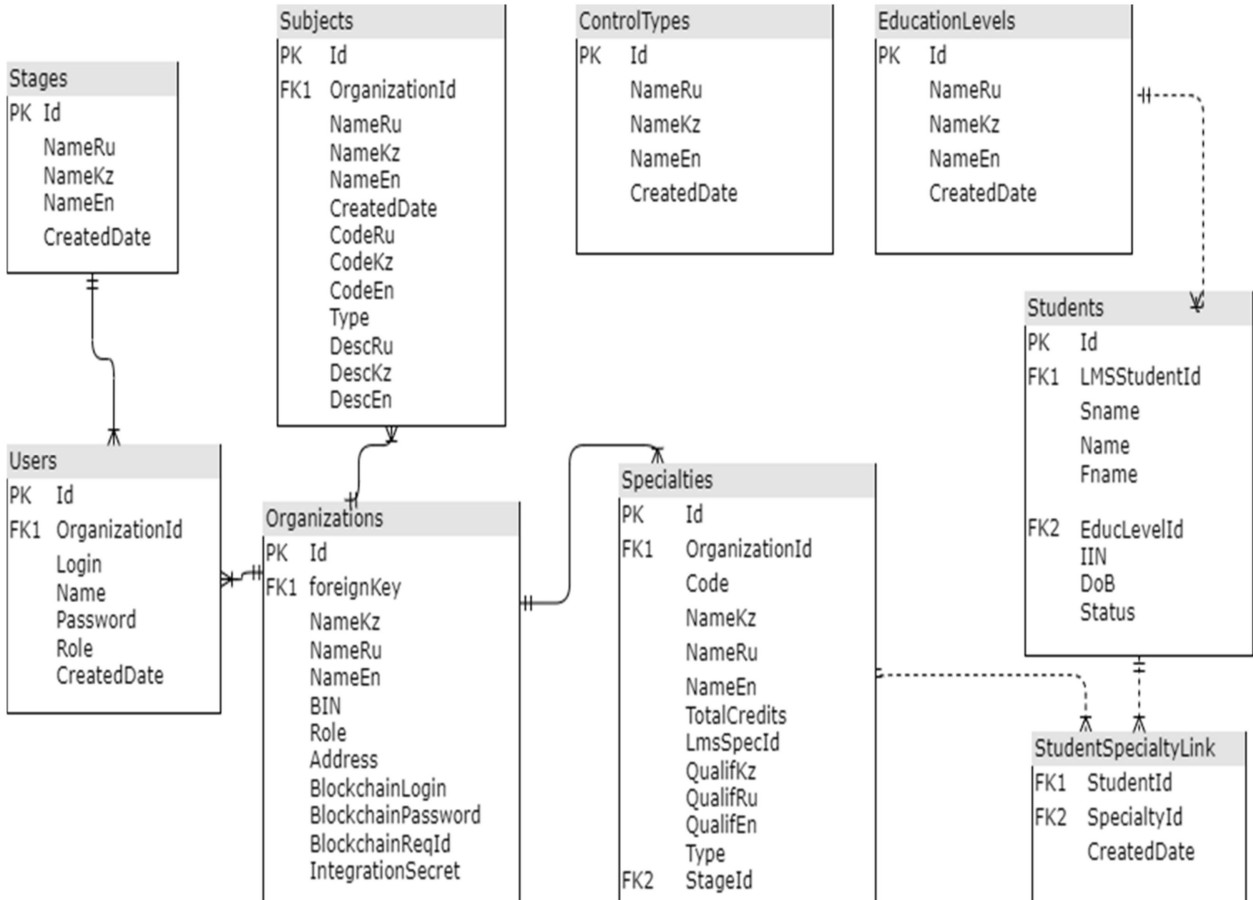

**Figure 4.** Off-chain database architecture.

The tables labeled Stages, ControlTypes and EducationLevels are constants. Universities should use their identifiers when working with other tables. Users and Organizations store information about platform participants with access rights. The Specialties (educational programs), Subjects (courses) and Students tables are filled in by the HEI participant when registering reference data, training and courses.

In the first stage, a record about the organization (university or examiner) is formed in this database, as shown in Figure 5. An account is created based on the results of the organization's registration. Then, the university (a new organization in the database) initiates the registration process of its educational programs. Moreover, the university must compare them with the corresponding training stages, which are constant in the database. The next step is for the university to register the entire compulsory and elective courses of this OP. In the last stage, the university registers its contingent of students in this database. The methods for registering reference data are organized in such a way that, when a record is added for all the listed entities, they return off-chain database identifiers, which can be stored in the learning management systems of universities. These identifiers serve as input information when making a transaction in the blockchain database.

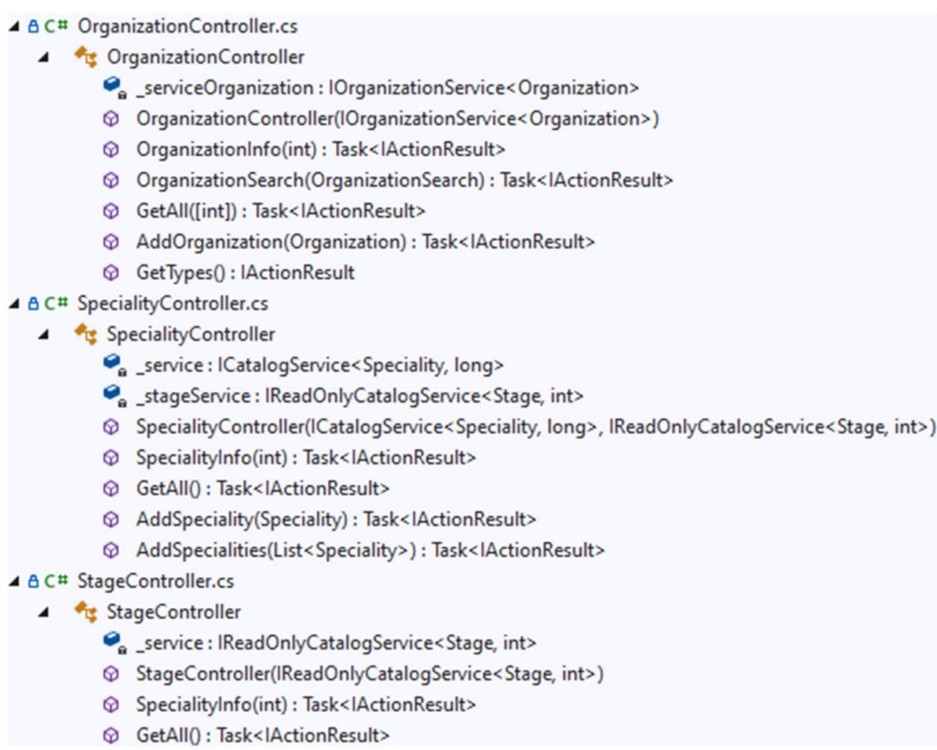

**Figure 5.** RestAPI methods for registering participant reference data.

The methods of the Blockchain category are designed to connect to the blockchain database and make a transaction on behalf of the university using its account from the off-chain storage and the blockchain address (Figure 6).

```
C# BlockOperationsController.cs
   BlockOperationsController
      _blockChainService : IBlockchainService
      _encryptionService : IEncryptionService
      BlockOperationsController(IBlockchainService, IEncryptionService)
      InsertDataIntoBlock(StudDataDto) : Task<IActionResult>
      GetStudentsFromBlock(int) : IActionResult
      GetStudentsFromBlockBySpecialityId(StudentSpecialityDto) : IActionResult
      CreateAccount(string) : IActionResult
```

**Figure 6.** RestAPI methods for making a transaction in the blockchain database.

For a complete description of the operation of the Blockchain category methods, it is necessary to explain the RabbitMQ technology used to unload request flows into the blockchain database. RabbitMQ is a software message broker [32]. Its main purpose is to receive, store, and send messages of binary data.

Storing data in a blockchain database consists of two stages:

Stage 1. The university calls the desired method and transmits information about the student, the educational program and the array of courses, with grades. The method is the message provider and queues the received information.

Stage 2. The service processes messages in the queue in the background, sending them to the blockchain database. If any errors occur, the service puts the message back in the queue for re-processing. If successful, a method published on the university side is called to transmit the block address and transaction status. This approach can allow several universities to send a large amount of information about their students simultaneously, without

worrying about losing an answer to a previously sent request. Additionally, sending data to the blockchain database in turn allows for uniform execution of all transactions.

The method for obtaining data from the blockchain database is as follows. The verifier calls to obtain the incoming parameters (student's IIN and organization's BIN), and then the method obtains the IDs of the student and the educational program of the organization from the corresponding tables of the off-chain database and accesses the blockchain database. Due to the fact that the student ID and the educational program with long types are stored as separate index fields in the block, the search is carried out fairly quickly.

## 4. Descriptors of the Second Criteria and Experimental Results

### 4.1. Transaction Costs

This section analyzes the transaction costs charged by the Ethereum blockchain consortium to launch the solution. First, the cost of a single smart contract, as described in the previous sections, was estimated. The next step was to estimate the costs of launching our solution in the Republic of Kazakhstan, namely the amount of gas that our solution consumes per year based on the number of students in all universities of the Republic of Kazakhstan.

The fee required for the Progress smart contract to be executed on the Ethereum network consortium is shown in Table 3. More precisely, the table shows how much gas was spent to launch the smart contract. To calculate these data from the smart contract code, we used Remix IDE. Alternatively, the Solidity compiler can be used.

**Table 3.** Prototype's gas cost.

| StudData | |
| --- | --- |
| **Function** | **Gas Used** |
| Constructor | 651,687 |
| addStudData | 550,000 |

The data in Table 3 indicate that the smart contract constructor is the most expensive function. The reason for this is that the input data for these constructors are larger than the input data for the other functions. Moreover, our designers initiate internal variables in non-volatile memory, which depends on expensive instructions of the Ethereum virtual machine. However, constructors are executed only once when deploying a smart contract. Thus, its cost is paid only once.

Let us turn to an assessment of how much gas is needed to use our solution in the Republic of Kazakhstan. Our estimate requires some approximate values, which are presented in Table 4.

**Table 4.** Approximate parameters for calculating cost.

| Description | Approximate Value |
| --- | --- |
| Active Kazakhstan HEIs | 122 |
| Average stages offered by an HEI | 3 |
| Average number of bachelors students entering HEIs per year | 144,139 |
| Average number of masters students entering HEIs per year | 17,308 |
| Average number of PhD students entering HEIs per year | 2304 |
| Average bachelor stage duration in years | 4 |
| Average master stage duration in years | 2 |
| Average PhD stage duration in years | 3 |
| Average bachelor stage courses required per semester | 6 |
| Average master stage courses required per semester | 5 |

According to the Review of the Higher Education System, in 2022, the number of higher education institutions amounted to 122. In most universities, students study at three levels: the levels of a bachelor's degree, master's degree and doctoral degree. Annually, more than 150,000 people begin their studies at the universities of the Republic of Kazakhstan.

We estimated approximately how many courses students would take and how long they would take to earn a degree. Note that this strongly depends on the degree that a student is pursuing. According to the order of the Minister of Education and Science of the Republic of Kazakhstan dated 20 April 2011, No. 152, 240 ECTS credits are required for a bachelor's degree; for a master's degree, it is 120; and for a doctoral degree, it is 180. The average durations of bachelors, masters and PhD studies are 4, 2 and 3 years, respectively.

Based on the above approximations, we could estimate the operating costs of our prototype in terms of gas for the entire higher education system of the Republic of Kazakhstan.

Figure 7 shows the costs during the first 4 years of deployment. We assumed a gradual increase, starting only with newly enrolled students. One transaction per semester is performed per student by a one-time calculation of the addStudData function. Depending on the student's degree, the function call is made from four to eight times per student. We excluded the costs of the initial deployment of powers under the smart contract, which tends toward zero, because, in the long term, these costs are distributed among all students and graduates enrolling at universities.

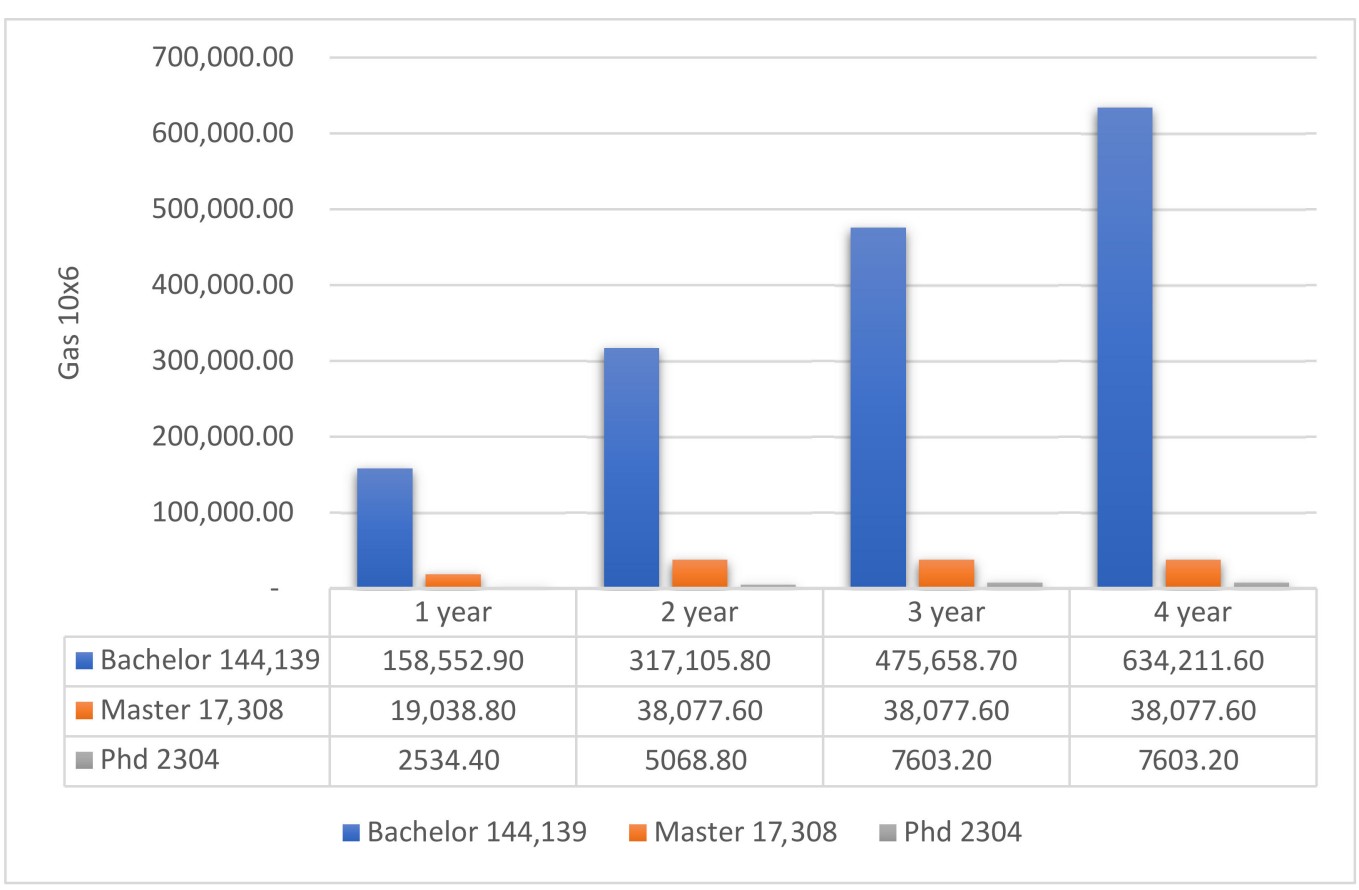

**Figure 7.** Estimated costs per year with gradual implementation for the state system of higher education of the Republic of Kazakhstan.

Comparing our solution with the processes of issuing paper diplomas and academic transcripts in terms of material and time costs is interesting. In fact, the cost of materials required for printing diplomas and that of human resources required for the preparation of these documents, as well as the time spent, are expected to be significant, but their accurate assessment is beyond the scope of this study. We are unable to calculate the costs of purchasing and printing diplomas because diploma forms are prepared at the level of the Ministry of Education and Science of the Republic of Kazakhstan and are provided to universities for free. However, we know that the official amount of time for the preparation of graduate education documents is 15 days [33]. This means that the employer must wait

for this time to receive a response to his request to the Ministry of Education and Science of the Republic of Kazakhstan regarding the verification of the authenticity of the graduate diploma, or the graduate must wait the same time to receive a duplicate document. The authentication or re-preparation of the document takes 15× X expenses, where X—daily staff maintenance expense in monetary terms. It considers the labor costs of each staff member involved in this process. In our solution, most of the time is spent on connecting to the network of the participant (employer or university) at the beginning, which can take a maximum of 3 days. Further operations to verify or obtain a duplicate do not require human costs and do not last more than 30 s in time. This is the average execution of a transaction for the preparation of a digital document of a student or graduate.

Our experiment shows that, for the execution of 1000 transactions, the blockchain consumes an average of 550,000 gas per transaction. Taking into account the number of semesters, the following expenses are incurred for each stage of study: bachelor's degree, 4.4 million gas; master's degree, 2.2 million gas; and doctoral degree, 3.3 million gas. Each semester, the platform processes a total transaction volume equal to the total number of university students.

### 4.2. Platform Performance in Terms of Speed and Volume of Transactions

This section discusses performance in terms of transaction execution speed, the amount of information stored in the consortium blockchain and network latency. The UniverCert platform is built on the principle of the Ethereum blockchain consortium. Compared with the public Ethereum blockchain, the role of the transaction is important here mainly for execution of the contract, not for the transfer of ETH cryptocurrency. Therefore, this article focuses on the performance and storage of executing Ethereum contracts.

In the Ethereum blockchain consortium, where authorized nodes are participants and smart contracts that can be assessed by the complexity of business logic are used, it is possible to calculate the necessary performance and storage volume to make technical decisions on network maintenance and node status. This is important because, with an increase in the volume of transactions, the performance of Ethereum decreases significantly, and a great deal of space is required for data storage. The authors of a measurement study proposed a method for predicting storage performance and growth by analyzing the relationship between transaction volume and the "state of the world" [34]. The "state of the world" is implemented using the "modified Merkle Patricia tree (trie)" (hereinafter MPT) [35]. The speed of transaction execution in Ethereum [36] is determined mainly by the operating time of the Ethereum Virtual Machine (EVM) and the modification of the "state of the world." In addition, the data volume increment depends on the scale of the transaction and the magnitude of the increment in the MPT tree. Consequently, an estimate of performance and memory size can be obtained, provided that the relationship between the volume of transactions and the "state of the world" is clarified. Taking into account the relative stability of EVM operation time and the predictability of contracts, we propose a formula for predicting transaction execution time and storage occupancy through the cost of maintaining the "state of the world." The MPT tree, or the "state of the world" of Ethereum, consists of two parts: the upper tree of the State Trie contracts and the lower Storage Trie, the root of which points to the contract.

Therefore, the average time to modify the MPT is determined as follows:

$$T_{mpt}(n) = \frac{t}{2} \log(an) \tag{1}$$

where $a$ is the number of contract deployments in the network, $n$ is the transaction volume, $t$ is the execution time for creating a node in the MTP tree (sha3() calculation and database access).

Given the specifics of the LevelDB database used by Ethereum [37], the time $t$ (access to the database) increases as the amount of data storage increases. However, the calculations use the average static value of $t$.

The next step is to calculate the average transaction execution time, as follows:

$$T_{avg}(n) = T_{exec} + T_{mpt}(n) \tag{2}$$

where $T_{exec}$ is the time to execute the contract on the virtual machine.

From the experimental results, we find that the average time of a database visit is t = 0.03 ms under the following conditions: one Ethereum node is used without a network, only one transaction is packed into a block and 1,000,000 transactions are randomly generated. The value of the Texec parameter for the blockchain consortium is relatively stable, and its impact on the total time can be neglected. Thus, the final formula for predicting the transaction execution time is as follows:

$$T_{predict}(n) = \frac{0.06 * \log(n)}{2} \tag{3}$$

Our experience is based on the execution of 1000 transactions with an average execution speed of ~4 s, as shown in Figure 8. Based on the speed of the transaction propagation over the public network (for the blockchain consortium, the average time should be even less), the total time to complete and confirm a transaction over the entire network is ~4.2 s. Using Formula (3), we can calculate the transaction execution time on larger transaction volumes.

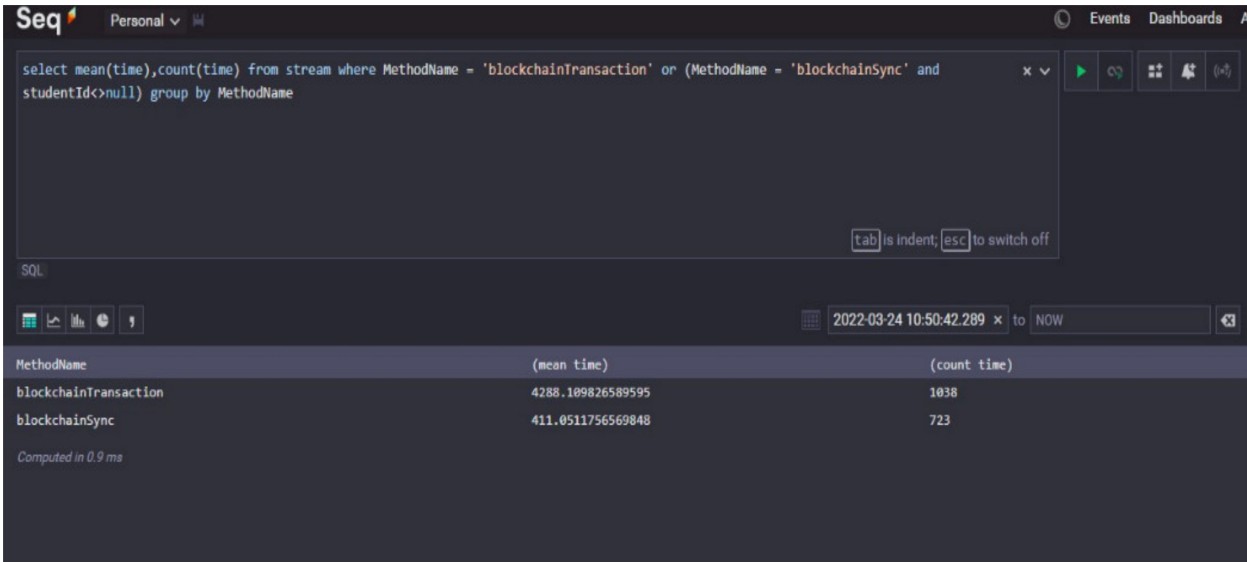

**Figure 8.** Average transaction execution time (blockchain transaction) when executing 1000 transactions.

### 4.3. Data Volume

This subsection analyzes the amount of data stored in the blockchain database after each transaction. In Table 3, we specify the addStudData function, which consumes no more than 550,000 gas. This is possible due to the amount of information. Let us take as a basis a bachelor student who studies up to six courses per semester, i.e., in one transaction, we need to put a record of six courses along with grades. Course data are specially serialized in the JSON format, which contains only critical data (course ID, grade, number of credits, semester and type of control). All data are digital. Our calculation shows that such a format of information for up to six records weighs no more than 1 KB. Table 5 shows the approximate amounts of data.

**Table 5.** Approximate data volumes.

| Course Data | Degree | Data Volume (KB) |
|---|---|---|
| Course ID Evaluation | Bachelor | 8 |
| Number of credits Semester | Master | 4 |
| Type of control | Doctoral | 6 |

In the article titled "Blockchain and smart contracts for registering higher education in Brazil," the authors proposed a formula for memory increment as a result of the modification of the MTP tree, which looks as follows:

$$S_{mpt}(n) = s_b \left( \frac{\log(an)}{2} - 1 \right) + s_1 + s_a + s'$$ (4)

where $s_b$ is the size of the filled branch, $s_1$ is the size of the end node, $s_a$ is the storage space occupied by the account state and $s'$ is the sum of storage trees for each account. If we include here the size of the $S_t$ transaction itself, then the final formula becomes

$$S_{sumavg}(n) = nS_t + \sum_{i=0}^{n} S_{mpt}(n)$$ (5)

From the experimental results of the authors, we find that, under the following conditions, when one Ethereum node is used without a network, only one transaction is packed into a block: one million transactions are randomly made, and the values of the parameters are $S_b = 564$ B, $S_l = 96$ B, $S_a = 102$ B and $S' = 0$. Here, the dimensions are not given in terms of the amount of memory in the disk, but in the form of the sum of the compressed key and the LevelDB value.

In our work, relatively, the same conditions are met, so we can use these parameter values in our calculations. Based on the fact that the total amount of information in one transaction does not exceed 1 KB, we can assume that the value of the parameter is $S_t = 8000$ B.

As a result, we can obtain Formula (6) for calculating the increase in the total storage volume of the blockchain database:

$$S_{predict}(n) = 564 * \left( \frac{\log(n)}{2} - 1 \right) + 96 + 102$$ (6)

The proposed data format is the right option for our solution in terms of optimizing transaction costs and increasing the efficiency of the platform in terms of transaction execution speed and minimizing data storage.

## 5. Conclusions

Documents attesting the history of the academic achievements of applicants confirming applicants' educational qualifications for the purpose of employment are increasingly becoming subject to fraud. There are many cases of forgery of official documents attesting educational credentials, as many universities still issue these documents on paper. Furthermore, requests for the confirmation of authenticity are burdensome in view of the fact that it is necessary to contact the university archive. In addition, there is no guarantee against the forgery of documents by unscrupulous employees. Therefore, in the context of the digitalization and automation of management processes in the higher education system, universities are increasingly paying attention to the integrity of data and their security and authenticity.

We offer a modern solution based on blockchain technology that can allow higher education institutions to provide full information to interested parties about academic performance and to verify the authenticity of documents.

The proposed UniverCert solution is based on the Ethereum blockchain consortium architecture, with a single Progress smart contract, which ensures that a university with the appropriate access level on the platform can record events only in relation to its students. The basic data about the student and the student's academic performance are stored in an off-chain database, and the identifiers of this database are used in UniverCert to execute transactions on the blockchain. This approach is taken to optimize transaction costs. We also present methods to obtain data from the blockchain database.

Based on the above analysis of transaction costs, we estimate the operating costs of our solution in terms of gas for the entire higher education system of the Republic of Kazakhstan.

The evaluation of the criteria for the functioning of the blockchain system shows that the speed of transaction execution in our solution allows us to quickly and efficiently identify a student and to transfer data on the student's academic performance. Moreover, the amount of information is small due to the translation of data into the JSON format and the use of an additional database.

The efficiency analysis shows that the developed solution has strong potential and competitive advantages for using blockchain technology for the entire higher education system of the Republic of Kazakhstan, with minimal costs.

**Author Contributions:** Conceptualization, Y.K. and G.M.; methodology, Y.K.; software, Y.K.; validation, Y.S., M.M. and G.M.; formal analysis, Z.S.; investigation, Y.K. and Y.S.; data curation, Y.K. and Y.S.; writing—original draft preparation, M.M., Z.S.; writing—review and editing, Y.K., Y.S. and G.M.; visualization, Y.K.; project administration, G.M. and Y.K.; funding acquisition, G.M. All authors have read and agreed to the published version of the manuscript.

**Funding:** This study was funded by the Science Committee of the Ministry of Education and Science of the Republic of Kazakhstan [Grant No. AP08857535-OT-20].

**Institutional Review Board Statement:** Not applicable.

**Informed Consent Statement:** Not applicable.

**Conflicts of Interest:** The authors declare no conflict of interest.

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
