# Peer review of "Ethereum-Based Information System for Digital Higher Education Registry and Verification of Student Achievement Documents"

_futureinternet, doi:10.3390/fi15010003_

Round 1

Reviewer 1 Report

A deeper review of English language is suggested, also by shortening long sentences. This action should make easier text reading and understanding.

Reviewer 2 Report

The authors present a distributed system for digital certification of university diplomas. The system is designed to be implemented in the Ethereum blockchain and  performance metrics related with the transaction costs are being studied.

Given the increase in the transaction fees that makes D-apps in the Ethereum blockchain to scale poorly, the authors should provide arguments regarding their choice of this network for the deployment of the smart contracts an why they didn't other EVM-compliant networks (e..g., Polygon or Solana) for their application. This could highly affect their overall results therefore I believe their thoughts on the mattes should be presented in the paper

Reviewer 3 Report

Overall paper is well written, and easy to read.

The education statistics of the Khazastan government do not seem to be referenced or official. It would be preferable to provide more details, to have a comparison with the current education system in terms of treatment costs, even if approximate, so that the reader can have a better cost-benefit perspective. Similarly, risks are listed, but not discussed quantitatively, even though they are part of the basic information in this document. They should be estimated to add realism to the study.

It is noted in the abstract that previous research generally ignores performance issues. There have been estimates for throughput, transaction time, and data memory usage. There is no summary table on these aspects. This table should list what is desired, and what has been simulated.

For the cost structure, there should be different breakdowns of costs, such as per student, per year, per study type (as done in the paper), per average university, and for the country.  Figure 7 is a repeat of Figure 6, not the costs as indicated in the title of Figure 7. It may provide some information on this subject.

Line 482, please add the subscript as shown in Equation 4 for s1, and sa.

There should be no commas at the end of the equations.

Please revise the references provided, they are a bit old.

Round 2

Reviewer 2 Report

The authors have addressed the reviewer's concerns. Even though the arguments could be better presented, their logic is clearly identified.

Some minor spelling mistakes should be taken care at the reviewed sentences. For example in line 234 the text writes "In general, the goals of the listed new networks well solve the shortcomings of our solution." while it should say "In general, the goals of the listed new networks will solve the shortcomings of our solution."

Author Response

Response to Reviewer 2 Comments

Point 1: The authors have addressed the reviewer's concerns. Even though the arguments could be better presented, their logic is clearly identified.

Some minor spelling mistakes should be taken care at the reviewed sentences. For example, in line 234 the text writes "In general, the goals of the listed new networks well solve the shortcomings of our solution." while it should say "In general, the goals of the listed new networks will solve the shortcomings of our solution."

Response 1:

Based on the revievwer’s comment, a spelling check was made as shown in the example above, and as a result, 2-3 such errors were corrected.

Reviewer 3 Report

Is the manuscript clear, relevant for the field and presented in a well-structured manner?

The paper could be improved, but content seems interesting.

 Are the cited references mostly recent publications (within the last 5 years) and relevant?

Some references are a bit old and could have been improved, like requested in the previous review.

Does it include an excessive number of self-citations?

Fine.

Is the manuscript scientifically sound and is the experimental design appropriate to test the hypothesis?

Yes, scientific soundness is sufficient for a short paper.

Are the manuscript’s results reproducible based on the details given in the methods section?

Yes.

Are the figures/tables/images/schemes appropriate? Do they properly show the data? Are they easy to interpret and understand?

This could be improved. The figure for the estimated costs is minimal.

Is the data interpreted appropriately and consistently throughout the manuscript? Please include details regarding the statistical analysis or data acquired from specific databases.

This could be improved. Paper is a bit poor on this aspect. Authors should give more details on their results.

Are the conclusions consistent with the evidence and arguments presented?

Fine.

Author Response

Response to Reviewer 3 Comments

Point 1:  Are the cited references mostly recent publications (within the last 5 years) and relevant?

Some references are a bit old and could have been improved, like requested in the previous review.

Response 1:

For the current work, the links provided contain reliable and up-to-date information. Blockchain technology is used specifically in the field of education. Thereofore, The amount of research on blockchain in education is not so much. despite of this, we took into account the recommendations of the reviewer, and made updates to some Some references, for example reference 5.

Point 2: Are the figures/tables/images/schemes appropriate? Do they properly show the data? Are they easy to interpret and understand?

Response 2:

We believe that we have given a sufficient number of figures to describe the costs of the transactions performed. We have shown in detail how much it is planned to spend in the context of one student, depending on the level of study and the period of study. We scaled these costs to the country level. Perhaps the cost values reflect the results of a small experiment. But the above formulas for calculating the transaction execution time and the speed of synchronization of network nodes show that with extended experiments, the results should not change much.

Point 3. Is the data interpreted appropriately and consistently throughout the manuscript? Please include details regarding the statistical analysis or data acquired from specific databases.

This could be improved. Paper is a bit poor on this aspect. Authors should give more details on their results.

Response 3:

We, the authors, within the framework of the research objectives, believe that we have given enough arguments and experimental results to describe the principles of our solution.
